# Cell Extracts Derived from Cypress and Cedar Show Antiviral Activity against Enveloped Viruses

**DOI:** 10.3390/microorganisms12091813

**Published:** 2024-09-02

**Authors:** Takashi Furukawa, Ayumu Inagaki, Takeshi Hatta, Suzuha Moroishi, Katsuki Kawanishi, Yuki Itoh, Shotaro Maehana, Mohan Amarasiri, Kazunari Sei

**Affiliations:** 1Department of Health Science, School of Allied Health Sciences, Kitasato University, 1-15-1, Kitasato, Sagamihara 252-0373, Japan; moroishi.suzuha@st.kitasato-u.ac.jp (S.M.); ah18215@st.kitasato-u.ac.jp (K.K.); ah19204@st.kitasato-u.ac.jp (Y.I.); ksei@kitasato-u.ac.jp (K.S.); 2Department of Mechanical Engineering, National Institute of Technology, Oita College, 1666 Maki, Oita 870-0152, Japan; a-inagaki@oita-ct.ac.jp; 3Department of Parasitology and Tropical Medicine, School of Medicine, Kitasato University, Sagamihara 252-0374, Japan; htakeshi@med.kitasato-u.ac.jp; 4Department of Medical Laboratory Sciences, School of Allied Health Sciences, Kitasato University, 1-15-1, Kitasato, Sagamihara 252-0373, Japan; smaehana@kitasato-u.ac.jp; 5Research Facility of Regenerative Medicine and Cell Design, School of Allied Health Sciences, Kitasato University, 1-15-1, Kitasato, Sagamihara 252-0373, Japan; 6Graduate School of Engineering, Tohoku University, 6-6-06, Aoba-Ku, Sendai 980-8579, Japan; mohan.amarasiri.b5@tohoku.ac.jp

**Keywords:** antiviral activity, bacteriophages, plant-derived cell extract, *Chamaecyparis obtusa*, *Cryptomeria japonica*, cold vacuum extraction

## Abstract

The antiviral efficacy of cell-extracts (CEs) derived from cypress (*Chamaecyparis obtusa* (Siebold & Zucc.) Endl., *C. obtusa*) and cedar (*Cryptomeria japonica* (Thunb. ex. L.) D.Don, *C. japonica*) was assessed using phi6 and MS2 bacteriophages, which are widely accepted surrogate models for enveloped and non-enveloped viruses, in order to verify their potential use as antiviral agents. Our results indicate that CEs derived from *C. obtusa* are dominantly composed of terpinen-4-ol (18.0%), α-terpinyl acetate (10.1%), bornyl acetate (9.7%), limonene (7.1%), and γ-terpinene (6.7%), while CEs derived from *C. japonica* are dominantly composed of terpinen-4-ol (48.0%) and α-pinene (15.9%), which exhibited robust antiviral activity against phi6 bacteriophage. Both CEs successfully inactivated the phi6 bacteriophage below the detection limit (10 PFU/mL) within a short exposure time of 30 s (log reduction value, LRV > 4). Through exposure experiments utilizing CEs with content ratios prepared via 2-fold serial dilutions (ranging from 3.13% to 100%), we demonstrated that the antiviral effect could be sustained up to a concentration of 25% (*C. obtusa* LRV = 3.8, *C. japonica* LRV > 4.3 at a 25% CE content ratio for each species). However, CEs with content ratios below 12.5% did not produce a significant reduction in bacteriophage concentration and consequently lost their antiviral effects. Conversely, both CEs did not exhibit antiviral activity against MS2 bacteriophage, a non-enveloped virus. Our findings reveal for the first time the potential of CEs derived from *C. obtusa* and *C. japonica* for use as antiviral agents specifically targeting enveloped viruses.

## 1. Introduction

Essential oils (EOs) contain intricate blends of lipophilic, low molecular weight, and volatile compounds, predominantly comprising terpenes (monoterpenes and sesquiterpenes), phenylpropenes, and their oxygenated derivatives (alcohols, esters, ketones, aldehydes, and phenols) [1,2,3]. More than 300 chemical compounds have been identified from plant extracts [4]. Typically, EOs consist of 20 to 60 chemical species presented in various concentrations, with the composition ratios depending on the plant type, plant age, chemotype, region and altitude where it is grown, and climatic conditions [1,5,6]. These EOs have been used primarily in perfumes, cosmetics, and food aromatics due to their high fragrance. Interestingly, their efficacy in controlling, safeguarding, and treating diverse human diseases has been established, and medical and pharmaceutical applications are also in focus [7,8,9,10]. Notably, numerous studies have highlighted the antibacterial [11,12], antifungal [13,14], and antiviral [15,16,17] activities of EOs and their compounds. Also, several review articles have summarized important information on the antiviral effects of EOs and their individual components derived from diverse plant species against a broad spectrum of viruses, including hepatitis virus, herpes virus, influenza virus, norovirus, and adenovirus [18,19,20]. Previous in vitro studies have demonstrated that the primary mechanisms underlying the antiviral and virucidal actions of EOs involve direct effects on extracellular viruses (virions), inhibition of virus binding, penetration, intracellular replication, and release from host cells [19,20].

The steam distillation and organic solvent extraction methods are widely utilized for extracting functional components, including EOs, from plant materials. While steam distillation extraction is a safe and traditional method for extracting EOs, it is performed at high temperatures, which can degrade heat-sensitive components in the plant. In contrast, the cold vacuum extraction method, a new extraction method, can obtain the functional components from plant cells with low heat tolerance since it is performed at a low temperature of 35–40 °C [21]. The liquid extract obtained using the above methods from plant cells is called “cell xtracts (CEs)”. Since CE also contains EO components, it has been reported to show antimicrobial activity similar to that of EOs [22,23,24].

Approximately two-thirds of Japan’s land area (250,500 km^2^) is covered by forests, of which approximately 102,000 km^2^ comprises planted forests [25]. Cypress (*Chamaecyparis obtusa* (Siebold & Zucc.) Endl., *C. obtusa*) and cedar (*Cryptomeria japonica* (Thunb. ex. L.) D.Don, *C. japonica*) make up around 70% of these planted forests and have been utilized as construction materials in Japan. Generally, during the processing of cypress and cedar wood for construction purposes, residual materials such as branches and leaves are generated, which hold significant value as woody biomass. During biomass processing of these residual materials, CEs derived from cedar and cypress are obtained. Previous studies have reported that EOs derived from *C. obtusa* and *C. japonica* exhibit antibacterial and antifungal activities similar to those of other EOs [26,27,28]. On the other hand, information regarding the antiviral activities of these CEs remains scarce. In this study, we evaluated the potential of CEs derived from *C. obtusa* and *C. japonica* as antiviral agents for enveloped and non-enveloped viruses using phi6 and MS2 bacteriophages as surrogates. Effects of treatment time and the CEs concentration on the bacteriophage inactivation were also elucidated.

## 2. Materials and Methods

### 2.1. CEs Derived from C. obtusa and C. japonica

CEs utilized in this study were generously supplied by Takafuji Co., Ltd. (Oita, Japan). CEs were extracted from the branches and leaves of cypress and cedar trees harvested in Oita prefecture, Japan, by using a batch-type cold vacuum drying system that did not involve the addition of water, developed by Takafuji Co., Ltd. (CE Extraction date: *C. obtusa*, 4/6/2022; *C. japonica*, 7/6/2022). Approximately 200 kg of the branches and leaves was loaded in the drying chamber. Then, the drying chamber was depressurized at −96 °C and kept at around 35 °C for 24 h by heating with hot water circulating around its perimeter. The evaporated liquid from the branches and leaves was cooled and collected as CE. Approximately 30 L of CE yield can be obtained from 200 kg of branches and leaves. These CEs were stored in the dark at room temperature until they were utilized in the antiviral efficacy experiments.

### 2.2. GC-MS Analysis

Both CEs were analyzed by headspace gas chromatography–mass spectrometry (GC–MS) analysis to identify their chemical compositions. Two milliliters of CE sample was transferred into a vial, and the vial was sealed. The headspace extraction was carried out for 5 min at 80 °C using a headspace autosampler (TriPlusTM RSH SMART Autosampler, Thermo Fisher Scientific, Waltham, MA, USA) with a 1:100 split ratio. The GC (TRACE1310, Thermo Fisher Scientific) was equipped with a DB-WAX (length: 30 m, diameter: 0.25 mm, film thickness: 0.25 μm, Agilent Technologies, Santa Clara, CA, USA) column. The heater temperatures of both the inlet and transfer lines were maintained at 250 °C. Helium was used as the carrier gas at a 1.0 mL/min flow rate. The column temperature was kept at 40 °C for 2 min, and it was programmed to 240 °C at a rate of 8 °C/min and kept constant at 240 °C for 5 min. The MS (ISQ LT, Thermo Fisher Scientific) was used in EI mode under the following conditions: ionization voltage, 70 eV; scan range, 30–550 m/z; ion source temperature, 280 °C. The chemical structure of each component was identified by comparing the mass data of their peaks with the standard library data, NIST Mass Spectral Search Program (Version 2.2).

### 2.3. Preparation of Bacteriophages phi6 and MS2 Stock Solutions

Preparation of bacteriophage stock solution was performed as previously described with slight modifications [29]. *Pseudomonas syringae* bacteriophage phi6 (phi6, NBRC 105899) and *Escherichia coli* bacteriophage MS2 (MS2, NBRC 102619) were selected as surrogates of enveloped and non-enveloped viruses, respectively. *P. syringae* (NBRC 14084) and *E. coli* (NBRC 13965) were used as host cells of phi6 and MS2 bacteriophages, respectively. The cultivation of *P. syringae* and *E. coli* involved the use of NBRC 802 and LB medium, respectively. To initiate infection, 150 µL of overnight (16–18 h) culture of host bacteria in broth medium and 50 µL of bacteriophage stock solution stored at −80 °C were combined in 1.5 mL tubes and gently mixed. Subsequently, the mixture (200 µL) was added to 4 mL of the top agar (0.6% agar concentration) in a 15 mL tube, and mixed gently. The top agar, containing bacteriophage and host bacteria, was then poured onto the bottom agar (1.5% agar concentration) in a Petri dish, followed by incubation at 30 °C for phi6 and 37 °C for MS2 overnight (20–24 h). After incubation, 4 mL of SM buffer was added to the agar plate where bacteriophage plaques had formed and incubated overnight (20–24 h) to resuspend the bacteriophage in the SM buffer. Subsequently, SM buffer containing bacteriophage was filtered using a 0.2 µm cellulose acetate filter to completely remove the host bacterial cells. This filtered bacteriophage suspension served as the bacteriophage stock solution, with its concentration determined through plaque assays (approximately 10^10^–10^11^ PFU/mL). The bacteriophage stock solution was stored at 4 °C until further utilization in the antiviral efficacy experiments.

### 2.4. Evaluation of the Antiviral Activity of CEs through Exposure Experiment

Evaluation of the antiviral activity of CEs was performed as previously described with slight modifications [30]. To conduct the exposure experiments, 1 mL of CE was added to a 1.5 mL tube, while 1 mL of SM buffer was used as a control in place of the CE. The bacteriophage stock solution was diluted to approximately 10^7^ PFU/mL using SM buffer. Subsequently, 10 µL of the bacteriophage solution was added to each tube containing the CE and control (final bacteriophage concentration of approximately 10^5^ PFU/mL), and the samples were incubated at specified times and temperatures (Figure 1 and Figure 2). To evaluate the impact of CE content ratios on antiviral activity, different content ratios of the CE ranging from 3.13 to 100% were prepared by 2-fold serial dilution using sterilized ultrapure water. Preliminary experiments confirmed that the concentrations of both bacteriophages were not changed when exposed only to sterile ultrapure water (as a negative control), which was used for dilution. Subsequently, 1 mL of each diluted CE was added to a 1.5 mL tube, and exposure experiments were conducted following the same method as described above. All CE exposure experiments were performed in triplicate. After incubation, the bacteriophage concentrations in the CE-exposed and control samples were determined by plaque assay. Samples predicted to have high bacteriophage concentrations were subjected to a 10-fold serial dilution using SM buffer and then analyzed by plaque assay. For example, a 10-fold diluted sample was prepared by mixing 900 µL of SM buffer and 100 µL of sample, and a 100-fold diluted sample was prepared by further diluting the 10-fold diluted sample 10-fold in the same manner.

### 2.5. Statistical Analysis

The Microsoft Excel 2019 software was employed for the statistical analysis of significant differences in bacteriophage concentration between the CE-exposed and control samples. Differences were deemed statistically significant when the *p*-value was less than 0.05.

## 3. Results

### 3.1. Chemical Composition of CEs

Table 1 shows the chemical compositions of CEs derived from *C. obtusa* and *C. japonica*. Components with a similarity index (SI) exceeding 800, as per the NIST library guidelines, were identified, while those with an SI below 800 are categorized under “Others” (Table 1). A distinct variation in the chemical components and their relative contents was observed between *C. obtusa* and *C. japonica*. The predominant constituents in the CE derived from *C. obtusa* were terpinen-4-ol (18.0%), α-terpinyl acetate (10.1%), bornyl acetate (9.7%), limonene (7.1%), and γ-terpinene (6.7%). Notably, terpinen-4-ol exhibited the highest relative content in *C. japonica* at 48.0%, followed by α-pinene at 15.9%.

### 3.2. Antiviral Activity of CEs against phi6 Bacteriophage

Figure 1 shows the antiviral effect of CEs derived from *C. obtusa* and *C. japonica* against phi6 bacteriophage. Initially, phi6 bacteriophage was exposed to each CE for 30 s, 5 min, and 1 h. Both CEs could inactivate phi6 bacteriophage to below the detection limit (10 PFU/mL) within 30 s of exposure. Consequently, the exposure time was standardized at 30 s, and exposure experiments were conducted using varying content ratios (100–3.13%) of each CE prepared through 2-fold serial dilutions. As for the CE derived from *C. obtusa*, the phi6 bacteriophage could be inactivated to below the detection limit at a CE content ratio of 50% or higher. At a CE content ratio of 25%, the antiviral effect was still observed; however, it was diminished, with approximately 1.16 ± 0.22 log PFU/mL of the bacteriophage remaining (LRV = 3.75). As for the CE derived from *C. japonica*, the phi6 bacteriophage could be inactivated to below the detection limit even at a CE content ratio of 25%. Conversely, no significant decrease in phi6 bacteriophage concentration was observed at a CE content ratio of 12.5% or less (*p* > 0.05). Following 1 h of exposure to the CEs derived from *C. obtusa* and *C. japonica* at 4 °C, the concentrations of phi6 bacteriophage were 2.33 ± 0.03 log PFU/mL and 1.45 ± 0.15 log PFU/mL, respectively, indicating a reduced antiviral effect compared to the exposure at 30 °C.

### 3.3. Antiviral Activity of CEs against MS2 Bacteriophage

Figure 2 shows the antiviral effect of CEs derived from *C. obtusa* and *C. japonica* against MS2 bacteriophage. The concentrations of MS2 bacteriophage in both the CE-exposed and control samples did not exhibit significant reduction (*p* > 0.05) at all exposure times, including the longest exposure time of 48 h, indicating the absence of antiviral effects by CEs derived from *C. obtusa* and *C. japonica* against the MS2 bacteriophage, a surrogate for non-enveloped virus. For this reason, the effects of CE content ratios and exposure temperature on the inactivation effect of MS2 bacteriophage were not determined.

## 4. Discussion

In our present study, we aimed to evaluate the antiviral potential of CEs derived from the branches and leaves of *C. obtusa* and *C. japonica*, which are traditionally utilized as construction materials in Japan, focusing on their activity against phi6 and MS2 bacteriophages. The main components of the CEs used in this study were terpinen-4-ol, α-terpinylacetate, bornyl acetate, limonene, and γ-terpinene for *C. obtusa* and terpinen-4-ol, and α-pinene for *C. japonica*. Both CEs exhibited remarkable antiviral effects against the phi6 bacteriophage, an enveloped virus, even with a short exposure time of only 30 s, resulting in an exceptionally high inactivation ratio (LRV > 4). On the contrary, no significant reduction in the concentrations of MS2 bacteriophage, a non-enveloped virus, was observed following exposure to the CEs. Our findings reveal for the first time that CEs derived from *C. obtusa* and *C. japonica* possess antiviral activities specifically targeting enveloped viruses, highlighting their potential effectiveness against pathogenic enveloped viruses such as SARS-CoV-2, influenza virus, and herpes virus. Exposure experiments using CEs prepared through 2-fold serial dilution (content ratio: 3.13–100%) demonstrated that the antiviral effects were evident at CE content ratios ranging from 25% to 100%. However, these effects were diminished when the content ratio dropped to 12.5% or below. This indicates that CEs below 12.5% do not provide a sufficient concentration of CE compounds and their constituents necessary for effective viral inactivation.

To date, the majority of studies investigating the antiviral activities of EOs have primarily focused on enveloped viruses, revealing significant antiviral effects against this viral group. Conversely, research on the antiviral activities of EOs against non-enveloped viruses is very limited, and the findings generally indicate minimal or no effect [20]. Our results also found that CEs derived from *C. obtusa* and *C. japonica* showed similar antiviral activities as EOs of other plant species. On the other hand, in recent decades, a few studies have identified specific EOs capable of inactivating non-enveloped viruses such as norovirus, adenovirus, rhinovirus, and M13 bacteriophages [31,32,33].

The variations in antiviral activity against enveloped and non-enveloped viruses are likely dependent on the composition of compounds derived from different plant sources and their respective ratios. Since it has been reported that the antimicrobial activity of EO is not solely contributed by any one component [34], the complexity of the EO compounds and their composition ratios can make it challenging to identify specific compounds responsible for antimicrobial activity. In this study, compounds such as terpinen-4-ol, α-terpinyl acetate, α-pinene, and limonene were detected as dominant components from the CEs derived from *C. obtusa* and *C. japonica*. These chemical compounds have also been detected from CEs derived from *C. obtusa* and *C. japonica* at high composition ratios in previous studies [1,35,36,37], and EOs containing these chemicals as major components have been reported to exhibit antibacterial and antifungal activities [38,39,40,41]. Especially, the EO extracted from tea tree (*Melaleuca alternifolia*) was reported to contain terpinen-4-ol as the main component [42,43,44,45], as were *C. obtusa* and *C. japonica* in the present study (Table 2). Several other predominant constituents were identified in these EOs and CEs, thereby establishing a notable similarity between *C. obtusa*, *C. japonica*, and tea tree-derived EOs and CEs. It was revealed that the combination of monoterpene alcohols obtained from *M. alternifolia* (CMA) showed noteworthy antiviral effect, ranging from 1 to 3 log reductions (0.0075% CMA, exposure time: 900 min) against the West Nile virus [45]. Li et al. reported that *M. alternifolia* concentrate significantly reduced the ability of influenza virus (H1N1) to infect host cells [44], and Garozzo et al. also reported that tea tree oil (TTO) had antiviral effects against influenza virus (H1N1) with an ID50 of 0.0006% of TTO [42]. While direct comparisons of the antiviral efficacy of these extracts with the aforementioned studies are difficult because of the differences in the extraction methodologies and the specific virus targeted, it is noteworthy that CEs derived from *C. obtusa* and *C. japonica* demonstrated high antiviral effects within a short exposure time. Interestingly, despite terpinen-4-ol being a prominent bioactive component effective against influenza virus [42,44], it was not effective against West Nile virus, although both of them are enveloped viruses [45]. Future investigations are desirable to elucidate the principal constituents for antiviral effects in CEs derived from *C. obtusa* and *C. japonica*, and their applicable agents.

In Japan, the utilization of biomass is being promoted from the perspectives of green innovation, Sound Material-Cycle Society, and carbon neutrality. In our study, the CEs were obtained during the biomass processing of residual materials, specifically branches and leaves of *C. obtusa* and *C. japonica*. Both CEs exhibited significant antiviral effects against phi6 bacteriophage, suggesting that these CEs may be considered as sustainable antiviral and/or virucidal solutions against pathogenic enveloped viruses.

## 5. Conclusions

In this study, we determined the chemical constituents and the antiviral activities of CEs derived from *C. obtusa* and *C. japonica* against phi6 and MS2 bacteriophages, serving as models for enveloped and non-enveloped viruses, respectively. This is the first report on the CEs derived from these plants by the cold vacuum extraction methods. A remarkable antiviral effect was observed against phi6 bacteriophage, an envelope virus, with complete inactivation achieved within only 30 s of exposure (LRV > 4). Furthermore, the antiviral effects of both CEs were observed within the CE content ratios of 25–100%, while lower content ratios showed reduced efficacy. As for MS2 bacteriophage, both CEs showed no significant decrease in bacteriophage concentration even after 48 h of exposure time, suggesting the absence of antiviral effects against non-enveloped viruses.

Based on these findings, we concluded that CEs derived from *C. obtusa* and *C. japonica* possess high application value as antiviral products such as sanitizer sprays targeting enveloped pathogenic viruses. On the contrary, we could not determine the lot-to-lot variation for the chemical compositions and antiviral activities of CEs. Therefore, we will address this point and target for commercialization of these CEs as an antiviral product in further studies.

## Figures and Tables

**Figure 1 microorganisms-12-01813-f001:**
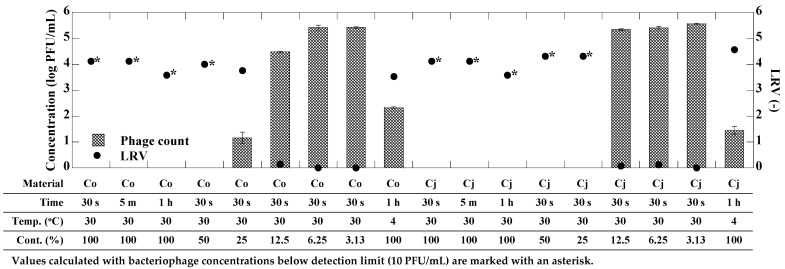
Concentration (log PFU/mL) and LRV (-) of phi6 bacteriophage upon exposure to each CE under several conditions. Co and Cj mean the CEs derived from *C. obtusa* and *C. japonica*, respectively. Error bars represent standard deviation of bacteriophage concentration.

**Figure 2 microorganisms-12-01813-f002:**
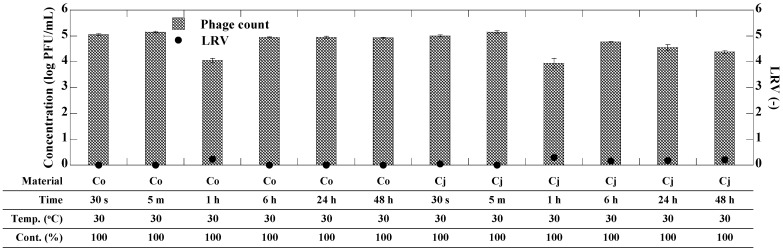
Concentration (log PFU/mL) and LRV (-) of MS2 bacteriophage upon exposure to each CE under several conditions. Co and Cj mean the CEs derived from *C. obtusa* and *C. japonica*, respectively. Error bars represent standard deviation of bacteriophage concentration.

**Table 1 microorganisms-12-01813-t001:** Components and their relative content of CEs derived from *C. obtusa* and *C. japonica*.

No.	Components	Retention Time (min)	*C. obtusa*		*C. japonica*	
			SI ^a^ (-)	Relative Content (%)	SI (-)	Relative Content (%)
1	α-Pinene	4.27	943	3.5	954	15.9
2	α-Thujene	4.36	816	0.4	851	0.6
3	(-)-β-Pinene	5.63	-	-	866	0.5
4	(1R)-α-Pinene	6.41	-	-	890	1.1
5	Myrcene	6.71	937	5.3	902	1.6
6	Terpinolene	6.97	919	2.0	905	1.2
7	Limonene	7.32	931	7.1	917	1.9
8	β-Phellandrene	7.49	850	0.5	846	0.5
9	Isoamyl alcohol	7.62	-	-	832	0.6
10	γ-Terpinene	8.19	946	6.7	931	3.0
11	*p*-Cymene	8.62	868	0.4	911	0.8
12	Terpinolene	8.85	942	2.3	907	0.8
13	1-Hexanol	10.16	878	0.9	-	-
14	*cis*-3-Hexen-1-ol	10.68	935	5.0	922	1.6
15	1-Octen-3-ol	11.75	930	2.1	891	0.6
16	Linalol	13.27	896	1.0	849	0.5
17	*trans*-*p*-Menth-2-en-1-ol	13.56	817	0.4	883	0.6
18	α-Himachalene	13.6	-	-	848	0.2
19	Bornyl acetate	13.8	949	9.7	923	1.7
20	Isobornyl acetate	13.88	864	0.6	-	-
21	Isocaryophillene	14.02	882	0.6	-	-
22	Cedrene	14.03	-	-	827	0.2
23	Terpinen-4-ol	14.15	930	18.0	933	48.0
24	Thujopsene	14.4	936	3.3	914	1.4
25	Cadina-3,5-diene	14.64	868	0.7	-	-
26	*cis*-Muurola-4(15),5-diene	15.12	906	1.2	-	-
27	α-Terpinyl acetate	15.5	939	10.1	921	2.3
28	(-)-Borneol	15.6	939	2.0	876	0.6
29	γ-Muurolene	15.69	870	1.1	857	0.5
30	α-Muurolene	15.88	900	0.4	848	0.2
31	α-Longipinene	16.05	886	0.4	836	0.2
32	δ-Cadinene	16.34	917	2.1	904	1.1
33	Elemol	20.52	919	0.8	925	1.1
34	Cedrol	21.02	821	0.3	821	0.3
35	γ-Eudesmol	21.59	893	0.6	893	0.7
36	τ-Muurolol	21.79	-	-	833	0.3
37	α-Eudesmol	22.2	868	0.5	889	0.6
38	β-Eudesmol	22.3	882	0.8	893	0.9
	Identified componetns (%)			90.7		92.1
	Others (%)			9.3		7.9
	Total (%)			100		100

^a^ SI: similarity index.

**Table 2 microorganisms-12-01813-t002:** Comparison of principal components and their relative contents of EOs derived from tea tree oils in previous studies and of CEs derived from *C. obtusa* and *C. japonica* in present study.

Oils		Tea Tree Oil	Tea Tree Oil	MAC ^a^	CMA ^b^	*C. obtusa*	*C. japonica*
Relative contents (%)	terpinen-4-ol	41.6	36.71	56–58	60.0–64.0	18.0	48.0
	γ-terpinene	21.5	22.20	20.65	0.5–1.0	6.7	3.0
	α-terpinene	10.0	10.10	9.8			
	p-Cymene	1.8	2.52		8.0–14.0	0.4	0.8
	α-terpineol	3.1	2.74		5.0–7.0		
	δ-Cadinene	1.0			4.0–6.0	2.1	1.1
	α-Terpinyl acetate					10.1	2.3
	Bornyl acetate					9.7	1.7
	Limonene	0.9				7.1	1.9
	α-Pinene	2.4				3.5	15.9
Reference		[43]	[42]	[44]	[45]	this study	this study

^a^ MAC: *Melaleuca alternifolia* concentrate. ^b^ CMA: combination of monoterpene alcohols obtained from *Melaleuca alternifolia*.

## Data Availability

The original contributions presented in the study are included in the article further inquiries can be directed to the corresponding author.

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
