# Peer review of "Cell Extracts Derived from Cypress and Cedar Show Antiviral Activity against Enveloped Viruses"

_microorganisms, 2024, doi:10.3390/microorganisms12091813_

Round 1
Reviewer 1 Report
Comments and Suggestions for Authors
The evaluation of the article was based on:
1. the research topic is very interesting, in the context of finding new sources of natural compounds, which reduce the degree of resistance of viruses to synthetic substances and thus reduce their degree of virulence and offer protection to the infected body;
2. the introduction presents data on essential oils, compounds recognized for their ability to annihilate the degree of virus infection of the human body; the purpose of the research and the objectives that will lead to the fulfillment of this objective are pointed out; the two resinous species studied, Chamaecyparis obtusa and Cryptomeria japonica, are part of the forests in certain areas of Japan; cypress and cedar are also recognized and cited in the literature and for the essential oil with multiple therapeutic virtues; it is interesting, however, that the authors want to demonstrate that the utilization of the biomass resulting from the processing of the wood of the two resinous species, the extraction of the volatile oil from them, can constitute an important source of natural compounds with antiviral action as well;
3. the materials and methods section is structured jointly; the analysis to obtain the essential oils, their GS-MS, applied in order to identify the dominant compounds in the oils is presented; preparation of stock solutions of bacteriophages, evaluation of antiviral activity;
4. are presented in detail at each stage of the research, comparisons obtained from the chromatographic analysis of the two types of essential oils are presented, their identification, as the authors mention, was done on the data entered in the spectra library; the following obtained after the determination of the antiviral actions are presented in the form of supporting graphs, these being correlated with the exposure time and the working temperature;
5. the discussions are extended, and the results obtained are correlated with the data from the specialized literature;
6. the conclusions are aimed at the results obtained and the research presented by the authors opens the premises of new directions in the evaluation of the specific actions of the essential oils obtained from a residual biomass of plant product.
Author Response
Thank you very much for reviewing and evaluating our manuscript. We have resubmitted a revised manuscript based on the comments of Reviewers 2 and 3. We believe the manuscript has been improved satisfactory.
Reviewer 2 Report
Comments and Suggestions for Authors
After a thorough review of the manuscript entitled “Cell-extracts derived from Chamaecyparis obtusa and Cryptomeria japonica show antiviral activity against enveloped viruses”, I highlighted some points that should be taken into consideration by the authors to improve the work.
1. Lines 24-26: “Our results indicate that CEs derived from C. obtusa and C. japonica dominantly composed of terpinen-4-ol (18.0%), α-terpinyl acetate (10.1%), bornyl acetate (9.7%), limonene (7.1%), and γ-terpinene (6.7%)”. Specify that these are major compounds of Chamaecyparis obtusa only.
2. Lines 42-46: The manuscript is about the antiviral activity of cell-extracts. I don't see the point in including an entire paragraph in the introduction talking about essential oils. Therefore, I suggest deleting the first paragraph.
3. Lines 66-75: In my opinion, this information should make up the first paragraph of the introduction to this manuscript. Additionally, authors should include further information on chemical constituents with antimicrobial effects previously identified in these two species.
4. Lines 82-85: Information such as voucher number, deposit in herbarium, and date of collection are extremely important in studies with medicinal plants.
5. Line 85: 200 kg? Is this information correct?
6. Lines 88-90: Were the extracts obtained not concentrated in a rotary evaporator or freeze dryer? How to determine the yield of the crude extracts?
7. Have the methodologies reported in sections “2.3. Preparation of bacteriophages phi6 and MS2 stock solutions” and “2.4. Evaluation of the antiviral activity of CEs through exposure experiment" been previously described in the literature? It is important to cite previous studies.
8. Line 136: Why was a conventional antiviral not used as a positive control to compare the results?
9. Line 142: Please provide more details on how these analyses were performed.
10. Lines 186-190: This information is very general and reads like an introduction/objective. I suggest deleting it. The discussion should be critical and focus specifically on the results obtained in this study.
11. Page 7. Lines 28-31: There is a lot of general information that can be left out of the discussion.
12. The entire discussion section needs to be reviewed and reworded by the authors.
13. Table 2: What is the scientific name of “tea tree oil”?
14. Conclusions. Considering the results obtained, what are the possible practical applications for the use of these extracts with antiviral potential? Authors should suggest in the conclusion the purposes for which these products can be used.
15. Page 8. Lines 82-84: This information is not clear. Table 1 shows a list of several compounds identified in both extracts. The antiviral effects may be related to the synergism between the different components of the extracts.
Author Response
Thank you very much for your helpful comments and reviewing our manuscript. We believe the manuscript has been improved satisfactory.
Comment 1: Lines 24-26: “Our results indicate that CEs derived from C. obtusa and C. japonica dominantly composed of terpinen-4-ol (18.0%), α-terpinyl acetate (10.1%), bornyl acetate (9.7%), limonene (7.1%), and γ-terpinene (6.7%)”. Specify that these are major compounds of Chamaecyparis obtusa only.
Response 1: We have revised the sentences as follows. (Page 1, Lines 23 – 27)
Our results indicate that CEs derived from C. obtusa dominantly composed of terpinen-4-ol (18.0%), α-terpinyl acetate (10.1%), bornyl acetate (9.7%), limonene (7.1%), and γ-terpinene (6.7%) while CEs derived from C. japonica dominantly composed of terpinen-4-ol (48.0%), and α-pinene (15.9%) which exhibited robust antiviral activity against phi6 bacteriophage.
Comment 2: Lines 42-46: The manuscript is about the antiviral activity of cell-extracts. I don't see the point in including an entire paragraph in the introduction talking about essential oils. Therefore, I suggest deleting the first paragraph.
Response 2: As the reviewer pointed out, the target of our manuscript is Cell-Extract (CE), but the main component of CE is essential oil, and as Table 1 shows, the CE constituents of C. obtusa and C. japonica in this study were almost identical to those of essential oils. The antiviral effects demonstrated by the CEs could be contributed to a large extent by the constituents contained in the essential oils. Therefore, we believe that previous knowledge and information on the chemical constituents in essential oils, their applications, and their antiviral effects are very important for this study. For these reasons, we would like to keep the paragraphs the reviewer pointed out in the manuscript. We much appreciate your understanding.
Comment 3: Lines 66-75: In my opinion, this information should make up the first paragraph of the introduction to this manuscript. Additionally, authors should include further information on chemical constituents with antimicrobial effects previously identified in these two species.
Response 3: Thank you for your suggestion. We considered moving the sentence the reviewer mentioned to the first paragraph, but moving it would require a modification of the overall sentences in Introduction. We have decided that the structure of the manuscript is sufficient for readers to understand the positioning, significance, and objectives of our study as it stands in the Original Manuscript. Unfortunately, to our knowledge, there is no available information on the specific antimicrobial activity of the chemical constituents in extracts (including essential oils) of C. obtusa and C. japonica. Thus, it is difficult to add further information. We much appreciate your understanding.
Comment 4: Lines 82-85: Information such as voucher number, deposit in herbarium, and date of collection are extremely important in studies with medicinal plants.
Response 4: CEs used in this study are not commercialized product and are extracts obtained as wastes during cold vacuum drying process of cypress and cedar branches and leaves to produce woody biomass. Therefore, there is no voucher number and deposit in herbarium of CEs. The dates of CE extraction from each residual material have been added as follows. (Page 2, Lines 89 – 90)
(CE Extraction date: C. obtusa, 4/6/2022; C. japonica, 7/6/2022)
Comment 5: Line 85: 200 kg? Is this information correct?
Response 5: 200 kg is correct.
Comment 6: Lines 88-90: Were the extracts obtained not concentrated in a rotary evaporator or freeze dryer? How to determine the yield of the crude extracts?
Response 6: CE was not concentrated. In this study, CE was defined as the evaporated water extracted from approximately 200 kg of cedar or cypress branches and leaves placed in a cold vacuum drying chamber owned by Takafuji Co., Ltd. Approximately 30 L of CE is extracted from approximately 200 kg of branches and leaves. This CE was provided by the company and used for antiviral experiments. We have added the following sentence regarding CE yield. (Page 2, Line 94)
A yield of approximately 30 L of CE can be obtained from 200 kg of branches and leaves.
Comment 7: Have the methodologies reported in sections “2.3. Preparation of bacteriophages phi6 and MS2 stock solutions” and “2.4. Evaluation of the antiviral activity of CEs through exposure experiment" been previously described in the literature? It is important to cite previous studies.
Response 7: Thank you for your suggestion. We have added the following references and sentences regarding the methods in 2.3 and 2.4, respectively.
Preparation of bacteriophage stock solution was performed as previously described with slight modifications [29]. (Page 3, Lines 114 – 115)
Evaluation of the antiviral activity of CEs was performed as previously described with slight modifications [30]. (Page 3, Lines 135 – 136)
29. Nagai, T.; Yamasaki, F. A Practical Manual for Handling Bacteriophages. MAFF Microorg. Genet. Resour. Man. 2019, 1–12. (in Reference)
30. Miyaoka, Y.; Kabir, M.H.; Hasan, M.A.; Yamaguchi, M.; Shoham, D.; Murakami, H.; Takehara, K. Virucidal Activity of Slightly Acidic Hypochlorous Acid Water toward Influenza Virus and Coronavirus with Tests Simulating Practical Usage. Virus Res. 2021, 297, 198383, doi:10.1016/J.VIRUSRES.2021.198383. (in Reference)
Comment 8: Line 136: Why was a conventional antiviral not used as a positive control to compare the results?
Response 8: The novelties of this study are 1. Cell-extracts (CEs) derived from C. obtusa and C. japonica by the cold vacuum extraction system contains similar chemical components with Essential oils (EOs) derived from C. obtusa and C. japonica, and 2. CEs exhibited remarkable antiviral effects against the phi6 bacteriophage, an enveloped virus, even with a short exposure time of only 30 seconds, resulting in an exceptionally high inactivation ratio (LRV > 4). These findings will be reported for the first time. We hope in the future to commercialize these CEs as antiviral products such as sanitizer sprays. Typical commercially available sanitizers use alcohol or hypochlorite, both of which are industrially produced and already have obvious antiviral effects. In contrast, since CE is a natural component, we considered that there was no significant meaning in comparing the antiviral activity of them. On the other hand, it is important to note that CE derived from cypress and cedar also showed antiviral effects similar to those of commercial alcohol and hypochlorite. We much appreciate your understanding.
Comment 9: Line 142: Please provide more details on how these analyses were performed.
Response 9: We have added the following explanations on the serial dilution method for enumerating bacteriophage concentration. (Page 4, Lines 152 – 154)
For example, a 10-fold diluted sample was prepared by mixing 900µL of SM buffer and 100µL of sample, and a 100-fold diluted sample was prepared by further diluting the 10-fold diluted sample 10-fold in the same manner.
Comment 10: Lines 186-190: This information is very general and reads like an introduction/objective. I suggest deleting it. The discussion should be critical and focus specifically on the results obtained in this study.
Response 10: Thank you for your suggestion. We have deleted the sentences.
Comment 11: Page 7. Lines 28-31: There is a lot of general information that can be left out of the discussion.
Response 11: We agree with the reviewer's comments. We have moved the sentences pointed out by the reviewer to the Introduction section because we believe they are important as general information on plant extracts. Thank you very much. (Page 1 – 2, Lines 44 – 47)
Comment 12: The entire discussion section needs to be reviewed and reworded by the authors.
Response 12: We have revised the manuscript according to the comments of Reviewers 1 and 2 and reconsidered the whole discussion. If there are any other discussion sections that need to be reconsidered, we would appreciate it if you could specify them.
Comment 13: Table 2: What is the scientific name of “tea tree oil”?
Response 13: The scientific name of tea tree is Melaleuca alternifolia. Has been added as follows. (Page 7, Line 34)
tea tree (Melaleuca alternifolia)
Comment 14: Conclusions. Considering the results obtained, what are the possible practical applications for the use of these extracts with antiviral potential? Authors should suggest in the conclusion the purposes for which these products can be used.
Response 14: Thank you for your suggestion. We are planning to further study the possibility of commercializing these CEs as a sanitizer spray with antiviral effects. We have added following sentence in the Conclusion section. (Page 8, Lines 72 – 74)
Based on these findings, we concluded that CEs derived from C. obtusa and C. japonica possess high application value as antiviral products such as sanitizer spray targeting enveloped pathogenic viruses.
Comment 15: Page 8. Lines 82-84: This information is not clear. Table 1 shows a list of several compounds identified in both extracts. The antiviral effects may be related to the synergism between the different components of the extracts.
Response 15: Thank you for your suggestion. We agree with the reviewer that the antiviral effect may depend on the synergistic effect of the various components contained in CE. On the other hand, since CE is naturally derived, the chemical composition and antiviral effect may vary from lot to lot. Therefore, we believe that it is necessary to clarify the differences in properties between lots in further research in order to commercialize CE as an antiviral product. We have revised the sentences as follows. (Page 8, Lines 74 – 77)
On the contrary, we could not determine the lot-to-lot variation for the chemical compositions and antiviral activities of CEs. Therefore, we will focus on this point and determined for commercialization as an antiviral product of CE in further researches.
Reviewer 3 Report
Comments and Suggestions for Authors
1. Both in title and in the first mention in abstract and main text the scientific names must be complete of authorship. Please see https://www.worldfloraonline.org/
2. "phenylpropanes" must be "phenylpropenes".
3. Line 57-60. The sentences are not clear enough.
4. Please, specify the yield of extraction.
5. Line 158: "Table 1. Table 1 Components.." should be "Table 1. Components.."
6. Minor english editing is needed. For example: "we determined the chemical constituents and their antiviral activities of CEs ..." should be "we determined the chemical constituents and the antiviral activities of CEs ..."
7. Line 28-29: "More than 300 chemical compounds have been identified from plant extracts [29]." the number should be higher. Perhaps, Do the authors mean the number of compounds in an EO?
Comments on the Quality of English LanguageMinor editing is needed
Author Response
Thank you very much for your helpful comments and reviewing our manuscript. We believe the manuscript has been improved satisfactory.
Comment 1: Both in title and in the first mention in abstract and main text the scientific names must be complete of authorship. Please see https://www.worldfloraonline.org/
Response 1: Thank you for your suggestion. We have revised in the title Chamaecyparis obtusa and Cryptomeria japonica to cypress and cedar, respectively. We have added the complete scientific names of cypress and cedar in the abstract and manuscript.
Cell-extracts derived from cypress and cedar show antiviral activity against enveloped viruses (Page 1, Lines 2 – 3)
The antiviral efficacy of cell-extracts (CEs) derived from cypress (Chamaecyparis obtusa (Siebold & Zucc.) Endl., C. obtusa) and cedar (Cryptomeria japonica (Thunb. ex. L.) D.Don, C. japonica) was assessed using phi6 and MS2 bacteriophages, which are widely accepted surrogate models for enveloped and non-enveloped viruses in order to verify their potential use as antiviral agents. (Page 1, Lines 20 – 23)
Cypress (Chamaecyparis obtusa (Siebold & Zucc.) Endl., C. obtusa) and cedar (Cryptomeria japonica (Thunb. ex. L.) D.Don, C. japonica) make up around 70% of these planted forests and have been utilized as construction materials in Japan. (Page 2, 70 – 73)
Comment 2: "phenylpropanes" must be "phenylpropenes".
Response 2: We have revised. Thank you very much. (Page 1, Line 43)
phenylpropenes
Comment 3: Line 57-60. The sentences are not clear enough.
Response 3: We have revised the sentences you pointed out as follows. (Page 2, Lines 60 – 68)
The steam distillation and organic solvent extraction methods are widely utilized for extracting functional components including EOs from plant materials. While the steam distillation extraction is a safe and traditional method for extracting EOs, it is performed at high temperatures, which can degrade heat-sensitive components in the plant. In contrast, cold vacuum extraction method, a new extraction method, can obtain the functional components from plant cells with low heat tolerance since it is performed at a low temperature of 35 – 40 °C [21]. The liquid extract obtained using above methods from plant cells is called “cell-extracts (CEs)”. Since CE also contains EO components, it has been reported to show antimicrobial activity similar to that of EOs [22–24].
Comment 4: Please, specify the yield of extraction.
Response 4: Approximately 30 L of CE is extracted from approximately 200 kg of branches and leaves placed in a cold vacuum drying chamber owned by Takafuji Co., Ltd. The CE yield has been added as follows. (Page 2, Lines 94)
A yield of approximately 30 L of CE can be obtained from 200 kg of branches and leaves.
Comment 5: Line 158: "Table 1. Table 1 Components.." should be "Table 1. Components.."
Response 5: We have revised. Thank you very much.
Comment 6: Minor english editing is needed. For example: "we determined the chemical constituents and their antiviral activities of CEs ..." should be "we determined the chemical constituents and the antiviral activities of CEs ..."
Response 6: Thank you for your suggestion. The English of the whole revised manuscript was carefully reviewed by all authors before resubmission. We hope English editing is sufficient and satisfactory.
Comment 7: "More than 300 chemical compounds have been identified from plant extracts [29]." the number should be higher. Perhaps, Do the authors mean the number of compounds in an EO?
Response 7: We believe that the information that more than 300 compounds have been detected in EO is correct, since it is not only in the Adams et al. [4] we referred to, but also in the following articles. We much appreciate your understanding.
Wińska, K.; Mączka, W.; Łyczko, J.; Grabarczyk, M.; Czubaszek, A.; Szumny, A. Essential Oils as Antimicrobial Agents—Myth or Real Alternative? Mol. 2019, Vol. 24, Page 2130 2019, 24, 2130, doi:10.3390/MOLECULES24112130.
Dhifi, W.; Bellili, S.; Jazi, S.; Bahloul, N.; Mnif, W. Essential Oils’ Chemical Characterization and Investigation of Some Biological Activities: A Critical Review. Med. (Basel, Switzerland) 2016, 3, 25, doi:10.3390/MEDICINES3040025.
Prusinowska, R.; Śmigielski, K.B. Composition, Biological Properties and Therapeutic Effects of Lavender (Lavandula Angustifolia L.). A Review. 2014, 60, doi:10.2478/hepo-2014-0010.
Round 2
Reviewer 2 Report
Comments and Suggestions for Authors
The authors responded to my comments and necessary changes were made to the manuscript.